# M-scan: A Multi-Scenario Causal-driven Adaptive Network for Recommendation

## ABSTRACT

We primarily focus on the field of multi-scenario recommendation, which poses a significant challenge in effectively leveraging data from different scenarios to enhance predictions in scenarios with limited data. Current mainstream efforts mainly center around innovative model network architectures, with the aim of enabling the network to implicitly acquire knowledge from diverse scenarios. However, the uncertainty of implicit learning in networks arises from the absence of explicit modeling, leading to not only difficulty in training but also incomplete user representation and suboptimal performance. Furthermore, through causal graph analysis, we have discovered that the scenario itself directly influences click behavior, yet existing approaches directly incorporate data from other scenarios during the training of the current scenario, leading to prediction biases when they directly utilize click behaviors from other scenarios to train models. To address these problems, we propose the **M**ulti-**S**cenario **C**ausal-driven **A**daptive **N**etwork (**M-scan**). This model incorporates a Scenario-Aware Co-Attention mechanism that explicitly extracts user interests from other scenarios that align with the current scenario. Additionally, it employs a Scenario Bias Eliminator module utilizing causal counterfactual inference to mitigate biases introduced by data from other scenarios. Extensive experiments on two public datasets demonstrate the efficacy of our M-scan compared to the existing baseline models.

## 1 INTRODUCTION

With the rapid development of e-commerce platforms, social networks, and other online services, recommendation systems [26] have emerged as crucial tools for personalized content recommendations, enhancing user experience, and increasing business revenue. Traditional recommendation algorithms, such as collaborative filtering [10, 27, 28, 33] and content-based filtering [1, 20, 35, 36], have been extensively employed and implemented. These recommendation algorithms rely on users' historical behavior data to train recommendation models that predict whether users will click on or like specific products.

However, traditional recommendation systems have limitations in dealing with complex scenarios and achieving precise predictions. In these systems, recommendation models are designed based on a single scenario, utilizing only the data available within that particular scenario for model training. While single-scenario models can capture variations in user behavior within the given scenario and make accurate predictions, they still encounter three main challenges: (1) Some scenarios suffer from data sparsity [29], especially in the case of cold-start scenarios [13]. (2) The absence of user information from other scenarios may lead to suboptimal performance and incomplete user representations. (3) Single-scenario models may result in resource waste, as large-scale commercial platforms often contain numerous scenarios such as multiple rankings and dozens of pages.

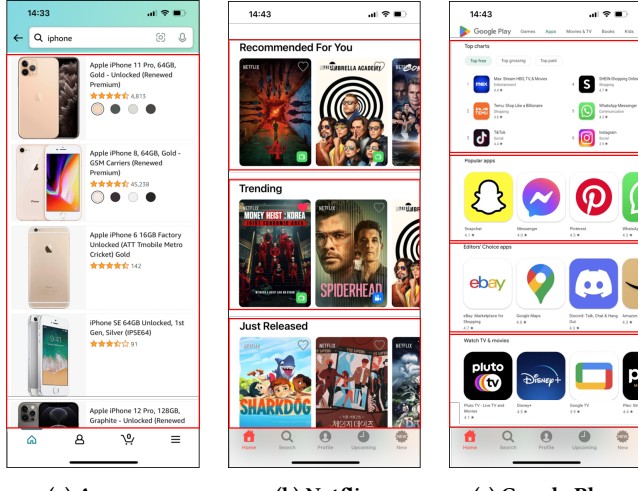

**(a) Amazon**    **(b) Netflix**    **(c) Google Play**

**Figure 1: Illustration of single-scenario and multi-scenario situations. Left: the whole page as a single-scenario in Amazon Shopping. Medium: horizontal lists as multi-scenarios in Netflix. Right: vertical and horizontal lists as multi-scenarios in Google Play Store**

To address the limitations of single-scenario models, the concept of multi-scenario modeling has been introduced. As is shown in Figure 1, Figure 1a shows a single-scenario situation, while Figure 1b and 1c depict two multi-scenario situations. Multi-scenario recommendation systems [2] integrate information from multiple scenarios through collaborative modeling, thereby enhancing the accuracy and robustness of recommendation algorithms. These systems leverage data from various scenarios to capture diverse user behaviors and train a unified model. This model simultaneously serves multiple scenarios and effectively mitigates resource waste.

In the current landscape of multi-scenario recommendation systems, previous works mostly focus on the model framework, such as MMOE [18] and STAR [32]. These approaches train models by incorporating data from various scenarios and designing specific network architectures tailored for multi-scenario settings. For instance, MMOE employs multiple expert networks to implicitly capture features from different scenarios, while STAR adopts a star-shaped topology network with separate sub-networks for each scenario. However, these existing approaches in multi-scenario modeling for recommendations have certain limitations. Firstly, they predominantly concentrate on designing the overall framework at the model architecture level, involving multiple networks or experts, while not explicitly modeling user interests at the individual user level, resulting in not only difficulty to train but also incomplete user representation and suboptimal performance. Secondly, they overlook the data biases introduced by the scenarios themselves, such as the impact of a scenario's location and size on user attention, which directly influences their likelihood to click on it.

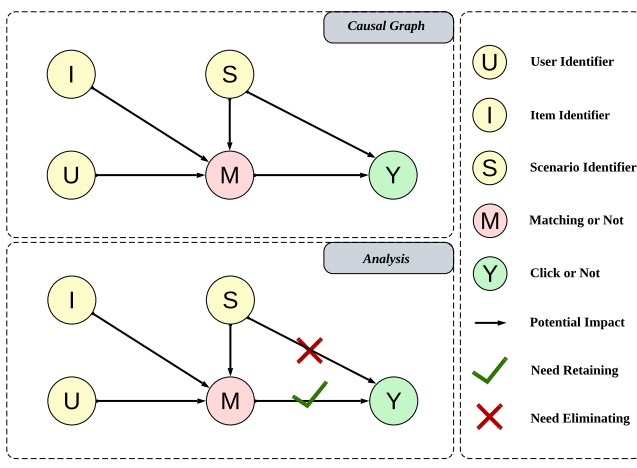

**Figure 2: Causal analysis of multi-scenario recommendation.**

Therefore, to gain a more intuitive and in-depth understanding of multi-scenario recommendation, we employ a causal graph [21] to depict the qualitative relationships among the variables involved. As is shown in Figure 2, each node represents a causal variable, and a directed edge $A \rightarrow B$ indicates that $A$ directly influences $B$.

- The causal graph consists of five nodes: $U, I, M, S,$ and $Y$. $U$ represents the user, $I$ represents the item, $M$ represents the matching degree between the user and the item, indicating the user's interest. $S$ represents the scenario, and $Y$ represents the click behavior. Both $U$ and $I$ naturally have an influence on the user's interest, denoted as $M$. Hence, there are edges $U \rightarrow M$ and $I \rightarrow M$ in the graph. Furthermore, the user's interest $M$ directly affects whether the user clicks on the item, resulting in the edge $M \rightarrow Y$.

- Next, we consider the impact of the scenario. The edge $S \rightarrow M$ indicates that different scenarios lead to distinct user interests. For example, the users' interests in the "gaming" scenario would differ from their interests in the "lifestyle" scenario.

- Finally, we have the edge $S \rightarrow Y$, suggesting that the scenario can directly influence the click outcome without affecting the user's interest. This is because factors like the location and size of the scenario can affect the user's field of view, subsequently influencing their click behavior. For instance, if Ranking A is large and positioned centrally, while Ranking B is small and located at the edge, Ranking A is more likely to be clicked because Ranking B may go unnoticed.

Based on the causal graph, we can observe that the influence of the scenario on the final click behavior can be categorized into two parts: $S \rightarrow Y$ and $S \rightarrow M \rightarrow Y$. When constructing multi-scenario recommendation systems, it is crucial to take both these influences into account. We not only need to gather insights into user interests and enhance user representations by utilizing data from various scenarios, but also need to address the biases introduced by different scenarios. For instance, if an item $I$ appears in scenario B, and B is less noticeable than other scenarios, then when incorporating data where the item $I$ is not clicked in scenario B, the model could mistakenly assume that the user genuinely dislikes $I$, disregarding the fact that the user simply didn't notice scenario B.

Presently, mainstream approaches to multi-scenario modeling encounter two primary issues: (1) They solely focus on the relationship $S \rightarrow M \rightarrow Y$ and overlook the direct influence of $S \rightarrow Y$.

(2) When considering $S \rightarrow M \rightarrow Y$, they focus on implicit model design, expecting the model to learn user interests in different scenarios, rather than explicitly modeling user interests. In practice, implicit modeling often requires a large number of parameters which poses difficulties in model training and parameter tuning. Moreover, the absence of explicit user interest modeling may cause incomplete user representations and suboptimal performances.

To address the aforementioned issues, we propose the **M**ulti-**S**cenario **C**ausal-driven **A**daptive **N**etwork (**M-scan**). M-scan incorporates two modules called **Scenario Bias Eliminator** and **Scenario-Aware Co-Attention** to address the two problems above respectively. (1) Scenario Bias Eliminator module models the direct influence of the scenario on click behavior and utilizes counterfactual causality to remove its effects. This ensures that our inference within the current scenario is not biased by other scenarios. (2) We want to use a widely successful attention mechanism [38] in order to extract explicit user interests from other scenarios. But unlike typical attention module [14, 15, 23, 47, 48] as target attention with candidate item as query or self-attention, M-scan introduces a specially designed co-attention [12, 17, 25] with current scenario's behavior also in the query. The Scenario-Aware Co-Attention module explicitly captures the impact of the scenario on user interests. It utilizes two user behavior sequences: the current scenario behavior sequence and the behavior sequences from all scenarios. By explicitly extracting user interests from other scenarios that align with the current scenario, it helps the model make better inferences.

The main contributions of our paper are summarized as follows:

- To the best of our knowledge, this is the first paper that analyzes the impact of scenarios not only on user interests but also directly on click behavior using causal graphs.

- We propose a novel model, M-scan, inspired by causal graphs. We design two modules to address two issues of multi-scenario modeling. The Scenario Bias Eliminator module eliminates the direct biases of other scenarios on click behavior. The Scenario-Aware Co-Attention mechanism explicitly models the impact of scenarios on user interests, extracting user interests from other scenarios that align with the patterns of the current scenario.

- We conduct offline experiments on two publicly available datasets and achieve promising results, demonstrating the effectiveness of our proposed model.

## 2 PRELIMINARIES

In this section, we will formulate the problem and then give a causal analysis of the multi-scenario recommendation problem.

## 2.1 Preliminaries

We provide a clear definition and formulation for the multi-scenario recommendation system, as well as define the current scenario user sequence and mixed scenario user sequence that we will use.

In the task of multi-scenario recommendation system, we have M users $\mathcal{U} = \{u_1, u_2, \ldots, u_M\}$, N items $\mathcal{I} = \{i_1, i_2, \ldots, i_N\}$, and P scenario numbers $\mathcal{S} = \{s_1, s_2, \ldots, s_P\}$.

We define a user interaction as a triplet that includes the user, item, and scenario, denoted by $y$ to represent the click or non-click. Therefore, the interaction records can be represented as a set

$$\mathcal{Y} = \{y_{uis}|u \in \mathcal{U}, i \in \mathcal{I}, s \in \mathcal{S}\}.$$

$$y_{uis} = \begin{cases} 1, & u \text{ has clicked } i \text{ in } s; \\ 0, & \text{otherwise.} \end{cases} \tag{1}$$

The multi-scenario recommendation model aims to provide an accurate prediction for $y_{uis}$ and obtain a recommendation list by ranking the scores of the candidate set. The predicted score $y_{uis}$ is derived from a model with parameters $\Theta$.

$$y_{uis} = \mathcal{F}_{\Theta}(u, i, s|\mathcal{H}_u) \tag{2}$$

where $\mathcal{H}_u = \{h_{b_1}, h_{b_2}, \ldots, h_{b_{N_{uh}}}\}$ represents the sequential behaviors of user u across all scenarios and there are $N_{uh}$ behaviors in all. It includes the common interests of the user in multiple scenarios. We can further obtain the scenario IDs for each behavior $h_{b_j}$, and $\mathcal{S}_u = \{s_{b_1}, s_{b_2}, \ldots, s_{b_{N_{us}}}\}$ indicates there are $N_{us}$ history behaviors in scenario s.

There is a notable challenge in this situation: during the final inference, it is incorrect to consider the user's interests across all scenarios; rather, we only need to focus on their interests in a specific scenario. Therefore, to achieve more precise predictions, we should adopt the following approach:

$$y_{uis} = \mathcal{F}_{\Theta}(u, i, s|\mathcal{M}_{us}) \tag{3}$$

Where $\mathcal{M}_{us}$ represents the user u's interest in scenario s. It is included in $\mathcal{H}_u$, and our goal is to extract it.

## 2.2 Causal-driven Analysis

Causal graphs are directed acyclic graphs [21] in which a node represents a variable and an edge represents the causal relationship between two variables. They are highly useful from a modeling perspective. In this section, we have built a causal-driven analysis for multi-scenario recommendation and gained inspiration for designing M-scan.

In the previous Figure 2, we presented the causal graph for the multi-scenario recommendation system. The following list defines the nodes and edges in the graph:

- **Node** $U$: A user identifier.
- **Node** $I$: An item identifier.
- **Node** $S$: A scenario identifier.
- **Node** $M$: The degree of matching between the user and the candidate item, indicating the user's interest and preference for the item.
- **Node** $Y$: The user's final click behavior on the item.
- **Edge** $U \rightarrow M$: The user's features influence the degree of matching between the user and the item.
- **Edge** $I \rightarrow M$: The item's features influence the degree of matching between the user and the item.
- **Edge** $S \rightarrow M$: The scenario in which the user and item are situated influences the degree of matching between them.
- **Edge** $M \rightarrow Y$: The degree of matching between the user and the item influences the likelihood of the user clicking on it.
- **Edge** $S \rightarrow Y$: The scenario features directly influence the likelihood of the user clicking.

In order to explore and model the impact of multiple scenarios, the most important edges that require significant attention are $S \rightarrow M \rightarrow Y$ and $S \rightarrow Y$. These edges represent the influence of

scenarios on click behavior through user interests and the direct impact of scenarios on click behavior, respectively. For instance, a user may exhibit different interests in the "Lifestyle" and "Gaming" scenarios, indicating the influence of $S \rightarrow M \rightarrow Y$. Moreover, if the "Gaming" category is in a small and remote slot, while the "Lifestyle" category is in a larger and more prominent slot, the "Lifestyle" category is more likely to be viewed and clicked, demonstrating the influence of $S \rightarrow Y$.

The presence of these two edges suggests that the click behavior Y in the multi-scenario recommendation system is influenced by both user interests and the scenarios themselves. However, during the inference process, we typically utilize intra-scenario inference, such as recommending 10 candidate items for a specific scenario. During inference, we should only consider user interests and exclude the influence of scenarios because the impact of $S \rightarrow Y$ remains the same for a particular scenario. Since the real data labels reflect the combined influence of all variables, directly supervising the model output with multi-scenario user interaction labels during training would introduce the bias of $S \rightarrow Y$, leading to biased or suboptimal predictions. Consequently, the model might mistakenly assume that the user's lack of clicks on items from other scenarios indicates disinterest, while in reality, it is simply due to the user's ignoring those items.

Hence, during training, it is crucial to model the influences of both $S \rightarrow M \rightarrow Y$ and $S \rightarrow Y$ in a specialized manner. During inference, the $S \rightarrow Y$ bias should be eliminated, with only the influence of $S \rightarrow M \rightarrow Y$ retained (as depicted in Figure 2). In the following sections, we will provide a comprehensive explanation of the M-scan model, which we have designed to capture the two aspects of scenario influence.

## 3 METHODOLOGY

In this section, we will give detailed descriptions of M-scan with its overview and the two primary designs, i.e., Scenario-Aware Co-Attention and Scenario Bias Eliminator.

### 3.1 M-scan Overview

When designing our model, we focus on two crucial aspects: eliminating biases from other scenarios and extracting user behaviors specifically for the current scenario from different scenarios. To address the bias issue, we employ a counterfactual causality approach and develop the Scenario Bias Eliminator module. To extract user behaviors for the current scenario explicitly, we introduce the Scenario-Aware Co-Attention mechanism. In this section, we will provide a comprehensive explanation of our M-scan framework on network architecture as well as its training and inference processes.

Figure 3 illustrates the overall framework of the M-scan model we designed. As is shown in the figure, M-scan takes five input components: user profile $u$, candidate item $i$, user behaviors in the current scenario $S_u$, historical behaviors across all scenarios $H_u$, and scenario features $s$. We start by feeding all inputs into the embedding layer to transform the sparse raw features $u, i, s, S_u, \mathcal{H}_u$ into low-dimensional dense embedding vectors $\mathbf{u}, \mathbf{i}, \mathbf{s}, \mathbf{S}_u, \mathbf{H}_u$.

In M-scan, our objective is to first model the user interest specifically for the current scenario and then leverage it to extract more

**Figure 3: Overall illustration of M-scan.**

similar interests from other scenarios. Therefore, the user behaviors for the current scenario, $S_u$, play a crucial role in the user representation, as the historical behaviors in the current scenario inherently contain the user interest representation in that scenario. So we feed $S_u$ into a scenario encoder, which can be implemented using Attention [48], GRU [7], transformer [37], or other encoders to model the representation of the current scenario behavior. We choose GRU here since scenario behaviors are in a sequential order. For the scenario encoder, the input consists of item embedding vectors $[s_{b_1}, s_{b_2}, \ldots, s_{b_{N_{us}}}]$. The hidden states $[h_1, h_2, \ldots, h_{N_{us}}]$ can be calculated using the following formula:

$$z_k = \sigma(\overline{W}_z s_{b_k} + \overline{U}_z h_{k-1} + \overline{b}_z)$$

$$r_k = \sigma(\overline{W}_r s_{b_k} + \overline{U}_r h_{k-1} + \overline{b}_r)$$

$$h_k = (1 - z_k) \odot h_{k-1} + z_k \odot \tanh(\overline{W}_h s_{b_k} + \overline{U}_h(r_k \odot h_{k-1}) + \overline{b}_h),$$
$$\tag{4}$$

where $\odot$ is the element-wise product operator, $\overline{W}$, $\overline{U}$, $\overline{b}$ are weight parameter matrices. .

With this, the historical behavior for the current scenario has been effectively modeled. Our next objective is to extract user interests that are similar to the current scenario from the behaviors observed in other scenarios. For this purpose, we utilize the historical behaviors across all scenarios $H_u$. To capture the interest of historical behaviors more accurately, we employ the widely adopted attention mechanism in sequential recommendation. Ideally, we would like to leverage only those historical behaviors that align with the interests of the current scenario. However, since we lack explicit knowledge about which behaviors from other scenarios are aligned with the current scenario's interests, we adopt an indirect measure to identify relevant historical behaviors.

Specifically, we quantify the alignment of interests by measuring the similarity between the behaviors of the current scenario and those of other scenarios. This is where our specially designed Scenario-Aware Co-Attention mechanism comes into play, and its detailed explanation will be provided in Section 3.2. Once we have computed the interest alignment $\beta_j$ for each historical behavior $h_{b_j}$, we employ a weighted aggregation approach using the attention mechanism to obtain the final representation of the user's history.

$$R_h = \sum_{j=1}^{N_{uh}} \beta_j \, h_{b_j}. \tag{5}$$

Next, the prediction $\hat{y}_m$ of user interest can be obtained by feeding all the features into a feed-forward neural network.

$$\hat{y}_m = \text{FFN}([u \oplus i \oplus s \oplus h_{N_{us}} \oplus R_h]), \tag{6}$$

Note that in $\hat{y}_m$, the subscript $m$ represents matching, which refers to the degree of matching between the user and the item, i.e., user interest. As discussed in Section 3.2, $\hat{y}_m$ represents the impact of user interest on the click behavior ($M \to Y$).

Additionally, there is another aspect of the scenario itself influencing the click behavior ($S \to Y$). The prediction $\hat{y}_s$ for this can be obtained simply by feeding the scenario feature $s$ into a feed-forward neural network.

$$\hat{y}_s = \text{Scenario FFN}(s), \tag{7}$$

*3.1.1 Training and inference process.* In Section 2.2, we conduct causal graph analysis and determine that the true sample labels $y$ are influenced by both $M \to Y$ and $S \to Y$. Hence, directly using a model trained with $y$ for inference would be inappropriate, requiring specific bias removal techniques. Consequently, we model these two influences separately, resulting in $\hat{y}_m$ and $\hat{y}_s$. In this section, we provide a summary of the model's training and inference processes. Detailed theoretical derivations and formulas will be presented in Section 3.3.

In the training process, since $y_m$ represents the influence of $M \to Y$ and $y_s$ represents the influence of $S \to Y$, we combine them in the Fusion layer using the following formula:

$$\hat{y}_{uis} = \hat{y}_m * \sigma(\hat{y}_s) \tag{8}$$

where $\sigma$ denotes the sigmoid function. Subsequently, we supervise the prediction $\hat{y}_{uis}$ using the true labels with the cross-entropy loss function:

$$\mathcal{L}_{uis} = \sum_{u,i,s \in \mathcal{D}} [-y_{uis} \cdot \log(\sigma(\hat{y}_{uis})) - (1 - y_{uis}) \cdot \log(1 - \sigma(\hat{y}_{uis}))].$$
$$\tag{9}$$

Similarly, we supervise the prediction $\hat{y}_s$, which solely focuses on the prediction of the scenario's impact on the click behavior:

$$\mathcal{L}_s = \sum_{u,i,s \in \mathcal{D}} [-y_s \cdot \log(\sigma(\hat{y}_s)) - (1 - y_s) \cdot \log(1 - \sigma(\hat{y}_s))]. \tag{10}$$

The final overall loss function is the weighted sum of the above two loss functions:

$$\mathcal{L}_{final} = \mathcal{L}_{uis} + \alpha \mathcal{L}_s \tag{11}$$

Here, $\alpha$ represents the balancing coefficient.

During the inference phase, we adopt the approach of scenario-specific prediction, where each scenario is predicted individually. Since the scenarios are the same, there is no variation in the influence of $S \rightarrow Y$. Therefore, we need to remove the previously modeled impact of $S \rightarrow Y$ to obtain an unbiased prediction that truly represents the user interest. The formula for this is as follows:

$$\hat{y}_{db} = \hat{y}_m * \sigma(\hat{y}_s) - c * \sigma(\hat{y}_s) \quad (12)$$

Here, $c$ is a hyper-parameter that represents the counterfactual reference state of $y_m$.

The theoretical analysis of this formula will be discussed in subsequent sections. Intuitively, this inference formula can be understood as an adjustment based on $y_{uis}$. For example, consider two scenarios $s_1$ and $s_2$, where $s_1$ is more prominent and likely to be clicked compared to $s_2$. In this case, if $y_{s_1} >> y_{s_2}$, subtracting the second term $c * y_s$ will make $y_{m_2}$, which was originally smaller than $y_{m_1}$, larger than $y_{m_1}$. This signifies that although the user is influenced by the scenario, they still clicked, which represents their genuine love for this item.

To provide a clear explanation of the training and inference processes, we have prepared pseudocode shown in Algorithm 1.

---

**Algorithm 1** M-scan

---

**Require:** user, item, scenario features
**Ensure:** $\hat{y}_{db}$(unbiased y)
    $\hat{y}_m = \mathcal{F}_1(user, item, scenario)$
    $\hat{y}_s = \mathcal{F}_2(scenario)$
   Training:
    $\hat{y}_{uis} = \hat{y}_m * \sigma(\hat{y}_s)$
    $\mathcal{L}_{uis} = \sum_{u,i,s \in \mathcal{D}} [-y_{uis} \cdot \log(\sigma(\hat{y}_{uis})) - (1 - y_{uis}) \cdot \log(1 - \sigma(\hat{y}_{uis}))]$
    $\mathcal{L}_s = \sum_{u,i,s \in \mathcal{D}} [-y_s \cdot \log(\sigma(\hat{y}_s)) - (1 - y_s) \cdot \log(1 - \sigma(\hat{y}_s))]$
    $\mathcal{L}_{final} = \mathcal{L}_{uis} + \alpha \mathcal{L}_s$
   Inference:
    $\hat{y}_{db} = \hat{y}_m * \sigma(\hat{y}_s) - c * \sigma(\hat{y}_s)$
   **return** $\hat{y}_{db}$

---

## 3.2 Scenario-Aware Co-Attention

In this section, we will introduce a module called the Scenario-Aware Co-Attention module, which is designed to extract user interests from other scenarios that are similar to the current scenario. The attention mechanism widely used in sequence behavior modeling is the target attention [23, 47, 48]. The key point of the target attention mechanism is to compute the relevance between items and each user behavior. The general formula is as follows:

$$\beta_j = \text{softmax}_j(\text{Attn}(\boldsymbol{h}_{b_j}, \boldsymbol{i})), \forall \boldsymbol{h}_{b_j} \in H_u \quad (13)$$

where $\text{softmax}_j$ represents the $j^{th}$ score in the softmax function, and $\boldsymbol{h}_{b_j}$ represents the $j^{th}$ behavior in the historical sequence $H_u$.

In traditional target attention mechanisms, it is assumed that all behaviors belong to the user's interest in the current scenario. However, in multi-scenario situations, such attention mechanisms are insufficient. We need to explicitly distinguish which behaviors can help represent interests in the current scenario and utilize them. The indicator of the current scenario's interests is the user's

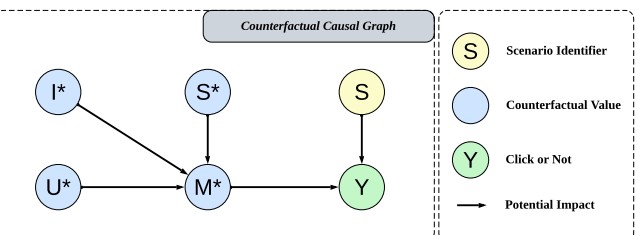

**Figure 4: Counterfactual causal graph of multi-scenario recommendation.**

historical behaviors of the current scenario. Therefore, we need to incorporate the historical behaviors of the current scenario into the attention mechanism to compute the relevance scores between behaviors from other scenarios and the current scenario's interest.

As is shown in the "Scenario-Aware Co-Attention" section of Figure 3, we calculate the Co-Attention matrix $C$:

$$C_{jk} = \text{Attn}(\boldsymbol{h}_{b_j}, \boldsymbol{i}, s_{b_k}), \ \forall \boldsymbol{h}_{b_j} \in H_u, \forall s_{b_k} \in S_u, \quad (14)$$

where $s_{b_k}$ represents the $k^{th}$ behavior in the historical sequence of the current scenario. "Attn" refers to a feed-forward neural network:

$$\text{Attn}(\boldsymbol{h}_{b_j}, \boldsymbol{i}, s_{b_k}) = \text{FFN}([\boldsymbol{h}_{b_j} \oplus \boldsymbol{i} \oplus s_{b_k}]), \quad (15)$$

where $\oplus$ represents the concatenation operator.

$C_{jk}$ denotes the relevance between the historical behavior $\boldsymbol{h}_{b_j}$, the current scenario behavior $s_{b_k}$, and candidate item $\boldsymbol{i}$. Next, we utilize a max-pooling layer to capture the most important correlations of $\boldsymbol{h}_{b_j}$ among $\{C_{jk}\}_{k=1}^{N_{us}}$. This signifies that as long as $\boldsymbol{h}_{b_j}$ is highly correlated with any item in the current scenario, it aligns with the interest distribution of the current scenario.

$$c_j = \text{MP}(\{C_{jk}\}_{k=1}^{N_{us}}). \quad (16)$$

Thus, $c_j$ represents the relevance of behavior $\boldsymbol{h}_{b_j}$ with the entire current scenario sequence $S_u$. Next, the attention score for $\boldsymbol{h}_{b_j}$ is computed using the softmax function.

$$\beta_j = \frac{\exp(c_j)}{\sum_{j'=1}^{N_{uh}} \exp(c_{j'})}, \quad (17)$$

And this will be used in Eq. (5). By doing so, we can extract historical behaviors from other scenarios that align with the interest of the current scenario and then utilize them.

## 3.3 Scenario Bias Eliminator

In Section 3.1, we have listed all the formulas and computations for this module, but we have not provided theoretical derivations and detailed explanations yet. In this section, we will provide a detailed theoretical derivation and explain its rationale.

### 3.3.1 Causal Counterfactual.
In the causal graph depicted in Figure 2, the variables in the graph influence each other, for example, both $M$ and $S$ have an impact on $Y$. Therefore, the variable $Y$ can be computed based on its ancestor nodes. Mathematically, this can be expressed as follows:

$$Y_{s,m} = Y(M = m, S = s) \quad (18)$$

where $Y(\cdot)$ represents the value function of $Y$. It can be observed that the causal graph provides explicit causal relationships, which aids us in constructing the architecture based on the directed edges.

Within the causal graph, a variable may have multiple influences on the next variable. These influences can be direct, such as $S \rightarrow Y$, or indirect, such as $S \rightarrow M \rightarrow Y$. We need to quantify these influences using mathematical formulas, and this is where counterfactual methods come into play. Counterfactual means considering S as s*, representing its removal from reality, for example, by setting S to be empty or an unknown constant value. Since s* is fixed or nonexistent, we can treat it as a reference status, having the same influence on other variables. Consequently, we can compute the total effect of all influences on $Y$ using counterfactual methods:

$$TE = Y_{S,M} - Y_{s^*,m^*} \qquad (19)$$

Here, M is also replaced by $m^*$ because $m^*$ also has an influence on Y. $m^*$ represents the counterfactual value of M.

*3.3.2 Scenario Bias Eliminator.* Furthermore, based on the causal graph structure, we can decompose the overall influence into two components: the impact of the scenario on the click behavior $S \rightarrow Y$, and the impact of user interest on the click behavior $M \rightarrow Y$. The influence of $S \rightarrow Y$ can be expressed as follows:

$$E_{S \rightarrow Y} = Y_{S,m^*} - Y_{s^*,m^*} \qquad (20)$$

where $Y_{S,m^*}$ represents the scenario $S$ obtained and only M is couterfactually removed.

We counterfactually remove M because we are only interested in calculating the influence of $S \rightarrow Y$ while disregarding $M \rightarrow Y$. The process of calculating the direct impact of $S$ on $Y$ is a causal counterfactual process since we cannot directly compute it using real data and can only rely on causal reasoning. Once TE and $E_{S \rightarrow Y}$ are calculated, $E_{M \rightarrow Y}$ can be obtained by subtracting the former from the latter.

$$\begin{aligned} E_{M \rightarrow Y} &= TE - E_{S \rightarrow Y} \\ &= (Y_{S,M} - Y_{s^*,m^*}) - (Y_{S,m^*} - Y_{s^*,m^*}) \\ &= Y_{S,M} - Y_{S,m^*} \\ &= Y(S = s, M = m) - Y(S = s, M = m^*) \end{aligned} \qquad (21)$$

At this point, we have obtained the calculated value for $E_{M \rightarrow Y}$, which represents the influence of $M \rightarrow Y$. Both terms, $Y(S = s, M = m^*)$ and $Y(S = s, M = m)$, can be fitted using neural networks. As mentioned in Section 3.1, we compute two prediction scores, $\hat{y}_m$ for $M \rightarrow Y$ and $\hat{y}_s$ for $S \rightarrow Y$. However, the ultimate target label should be $Y(S = s, M = m)$, representing the actual click behavior. Therefore, during training, to reconstruct the true click behavior, we combine these two branches together, which is shown in Eq. (8).

During inference, since the evaluation criterion is based on individual scenarios, the impact of scenarios on click behavior remains consistent. Therefore, we perform inference solely based on user interest, which is represented by the previously calculated $E_{M \rightarrow Y}$ in Eq. (12), as it has already removed the influence of $S \rightarrow Y$.

## 4 EXPERIMENTS

In this section, we show the experimental settings and results. Three research questions lead the following discussions, and our implementation code of M-scan is publicly available[1].

- **RQ1:** Does M-scan outperform existing multi-scenario recommendation models?

---
[1]https://anonymous.4open.science/r/M-scan-9B48

- **RQ2:** Are both innovative modules of M-scan effective?
- **RQ3:** What is the impact of the balance coefficient $\alpha$ and the hyperparameter $c$ on the results?

### 4.1 Experimental Settings

We conduct experiments using two publicly available multi-scenario datasets. The descriptions and statistics of the two datasets are detailed in appendix A.1.

*4.1.1 Baselines and Hyper-parameters.* To demonstrate the effectiveness of our proposed model, we compare it with different state-of-the-art models: Single, Mixing, Finetune, Shared Bottom, MMOE, PLE, AESM2, and M2M. The details of these models and the hyper-parameters are introduced in appendix A.2.

*4.1.2 Evaluation Metric.* We both evaluate the performance of the models in a single scenario and in all the scenarios. The evaluation metric used are the commonly used **AUC**(Area Under the Curve) [6, 9] and **RelaImpr** [31, 43] to the **Single** model. Since model **Single** cannot be tested in all the scenarios, **RelaImpr** is calculated by **Mix** in row "#All" in Table 1.

### 4.2 Overall Performance

The performance of the proposed M-scan and other baselines are presented in Table 1. #All means the whole dataset with all the scenarios. #1,#2,#3 represent three scenarios respectively. In Cloud Theme, we randomly choose 3 scenarios in 355 scenarios to demonstrate in the table. Model **Single** and **Finetune** don't have results in row #All because they can only be tested in one scenario. We have the following observations:

- M-scan outperforms the state-of-the-art baselines in two datasets significantly with p-value < 0.05 against the best baseline. This demonstrates that our M-scan model effectively captures the distribution within each scenario and accurately predicts user interests. Additionally, there are significant improvements in AUC in each scenario, which shows the robustness of our M-scan within different scenarios. The reason why the significance level does not drop below 0.05 in certain scenarios is probably attributed to the limited data within these scenarios.
- In the whole Aliccp dataset, the PLE model achieves the best performance among the baselines. On the other hand, the MMOE model, which also uses a multi-expert framework, performs significantly worse than PLE. This suggests that the approach of dividing the expert networks into shared and scenario-specific parts in PLE is effective, highlighting the specific characteristics of multi-scenario data distributions.
- While AESM2's overall performance is even worse than MMOE, it performs on par with PLE in certain scenarios. This is because AESM2 automatically selects experts as shared or scenario-specific experts. Despite this clever approach, it requires accurate representations of all scenarios. If a scenario representation is inaccurate, selecting unreliable experts can lead to poor results. Therefore, AESM2 only performs well in certain scenarios.
- In the Cloud Theme, M2M achieves the best overall performance over the baselines. However, it doesn't perform so well in Aliccp. This could be due to its unique meta-learning mechanism. Compared to datasets with only three scenarios, the dataset with 355

**Table 1: Performance comparison against baselines. The best result for all the models is given in bold, while the second-best is underlined. ∗ represents the significance level p-value < 0.05 against the best baseline.**

| Dataset | Scenario | Single | | Mix | | Shared bottom | | Finetune | | MMOE | | PLE | | AESM2 | | M2M | | M-scan | |
|---|---|---|---|---|---|---|---|---|---|---|---|---|---|---|---|---|---|---|---|
| | | AUC | RelImp | AUC | RelImp | AUC | RelImp | AUC | RelImp | AUC | RelImpr | AUC | RelImp | AUC | RelImpr | AUC | RelImp | AUC | RelImp |
| Aliccp | #1 | 0.6557 | - | 0.6634 | 1.17% | 0.6667 | 1.68% | 0.6699 | 1.71% | 0.6606 | 0.75% | 0.6672 | 1.75% | 0.6644 | 1.33% | 0.6523 | -0.52% | **0.6782**∗ | **3.43%** |
| | #2 | 0.6514 | - | 0.6575 | 0.94% | 0.6589 | 1.15% | 0.6647 | 2.04% | 0.6550 | 0.55% | 0.6633 | 1.83% | 0.6385 | -1.98% | 0.6430 | -1.29% | **0.6685**∗ | **2.63%** |
| | #3 | 0.6061 | - | 0.6361 | 4.95% | 0.6313 | 4.16% | 0.6321 | 4.29% | 0.6304 | 4.01% | 0.6405 | 5.68% | 0.6216 | 2.56% | 0.6226 | 2.72% | **0.6513**∗ | **7.46%** |
| | #All | - | - | 0.6574 | - | 0.6573 | -0.02% | - | - | 0.6543 | -0.47% | 0.6621 | 0.71% | 0.6420 | -2.34% | 0.6441 | -2.02% | **0.6714**∗ | **2.13%** |
| ĊloudTheme | #1 | 0.7769 | - | 0.8037 | 3.45% | 0.8070 | 3.87% | 0.8099 | 4.25% | 0.8099 | 4.25% | 0.8117 | 4.48% | 0.8063 | 3.78% | 0.8063 | 3.78% | **0.8120** | **4.52%** |
| | #2 | 0.7611 | - | 0.8011 | 5.12% | 0.8006 | 5.19% | 0.8032 | 5.53% | 0.8013 | 5.28% | 0.8031 | 5.52% | 0.8010 | 5.24% | 0.8010 | 5.24% | **0.8070**∗ | **6.03%** |
| | #3 | 0.7287 | - | 0.7738 | 6.19% | 0.7721 | 5.96% | 0.7754 | 6.41% | 0.7728 | 6.05% | 0.7756 | 6.43% | 0.7746 | 6.30% | 0.7744 | 6.27% | **0.7757** | **6.45%** |
| | #All | - | - | 0.7536 | - | 0.7543 | 0.09% | - | - | 0.7553 | 0.23% | 0.7563 | 0.36% | 0.7551 | 0.20% | 0.7570 | 0.45% | **0.7608**∗ | **0.96%** |

scenarios is better suited for adjusting network weights based on different scenarios. Because it's almost impossible to use 355 experts in the network when using MMOE-based models. So when it comes to a large number of scenarios, it is more feasible to model them in one network together rather than individually(using experts). This is also one of the advantages of M-scan.

- Most importantly, we observe that Finetune outperforms other baselines in certain scenarios. It's probably because Finetune uses behaviors in the current scenario to train a model in the $2^{nd}$ stage, reminding the model of its original intention. It inspires us that in the field of multi-scenario recommendation, further improving model architecture has become increasingly complex and hard to train but has only marginal or even negative improvements due to excessive parameters or high requirements for representation. Therefore, rather than implicitly learn unique representations of multiple scenarios through model architecture, we believe it more effective to explicitly model the unique interest representations of multiple scenarios at feature or data level, reminding the model of its original intention, a.k.a current scenario, which is also one of the main contributions of M-scan.

## 4.3 Ablation Study(RQ2)

In this section, we conduct ablation experiments to analyze the effectiveness of two components in M-scan. The main modules of M-scan are the Scenario-Aware Co-Attention (SACA) and the Scenario Bias Eliminator (SBE). Next, we examine the performance of two sub-models:

- w/ SACA, w/ SBE: This is the complete version of M-scan, as depicted in Figure 3
- w/ SACA, w/o SBE: This is a sub-model of M-scan without the Scenario Bias Eliminator module. It neither involves causal reasoning nor includes $\hat{y}_s$. Instead, it directly trains $\hat{y}_m$ calculated by the network using the real data labels and uses $\hat{y}_m$ for inference.
- w/o SACA, w/ SBE: This is a sub-model of M-scan without the Scenario-Aware Co-Attention module. It removes the component that extracts the current scenario user interests from historical behaviors of other scenarios.
- w/o SACA, w/o SBE: This is a sub-model of M-scan without both the Scenario Bias Eliminator and the Scenario-Aware Co-Attention modules. Similar to other mainstream state-of-the-art models, it only utilizes the historical behaviors from the current scenario as features.

The results are shown in Table 2. we can observe that the performance of M-scan significantly decreases when the *SBE* module is removed. This finding confirms the existence of scenario bias in the field of multi-scenario recommendation. The Scenario Bias

**Table 2: The AUC of M-scan and sub-models in ablation study**

| M-scan SACA | SBE | Aliccp | Cloud Theme |
|---|---|---|---|
| ✓ | ✓ | 0.6714 | 0.7608 |
| ✓ | ✗ | 0.6671 | 0.7581 |
| ✗ | ✓ | 0.6618 | 0.7591 |
| ✗ | ✗ | 0.6573 | 0.7477 |

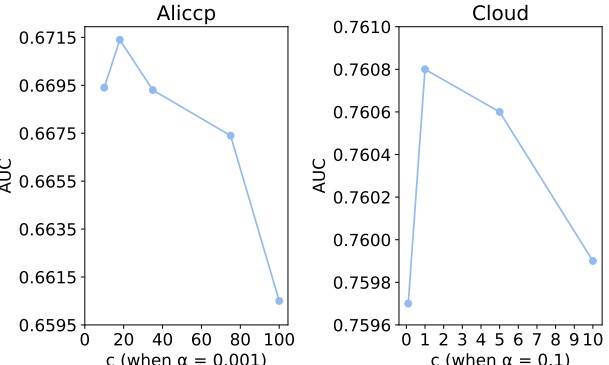

**Figure 5: Performance of M-scan using different $c$ values of Eq. (12) on two datasets.**

Eliminator we design effectively mitigates the scenario bias and leads to improved performance.

Additionally, we can observe that the performance of M-scan also significantly decreases when the SACA module is removed. This finding confirms the importance of incorporating the interests from other scenarios as features into the neural network. The Scenario-Aware Co-Attention module effectively extracts user interests from other scenarios that align with the current scenario.

Finally, when both the SBE and SACA modules are removed, the performance further deteriorates. This indicates that our SACA module and SBE module are both crucial.

## 4.4 Hyperparameter Study(RQ3)

In this section, we conducted hyperparameter experiments on the balance coefficient $\alpha$ and the counterfactual hyperparameter $c$. The results are shown in Figure 5 and Figure 6.

In Figure 5, we can observe that the evaluation metric AUC initially increases and then decreases with the increase of the counterfactual hyperparameter $c$, reaching a maximum value. The experiment demonstrates that moderation is the key when determining the amount of counterfactual bias removal represented by $c$. If $c$ is

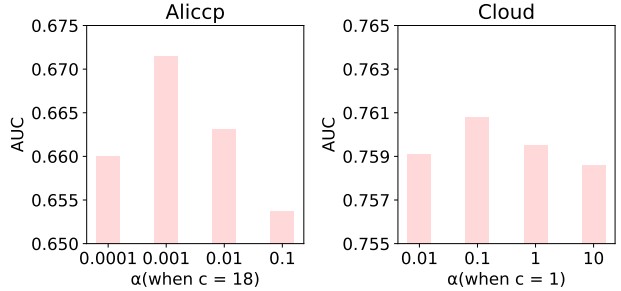

**Figure 6: Performance of M-scan using different $\alpha$ values of Eq.** (11) **on two datasets.**

too large, the bias is excessively removed, and if $c$ is too small, the bias is not adequately removed, leading to suboptimal performance.

In Figure 6, we can also see that the evaluation metric AUC initially increases and then decreases with the increase of the balance coefficient $\alpha$, reaching a maximum value. The experiment verifies the existence of a balance between the primary cross-entropy function $L_{uis}$ and the secondary cross-entropy function $L_s$. When $\alpha$ approaches 0, $L_s$ does not exist, resulting in biased outcomes. Conversely, when $\alpha$ approaches infinity, $L_{uis}$ ceases to exist, impeding the proper training of the recommendation model.

From the Figure 5 and 6, we can observe some patterns in hyperparameters adjusting. Since Aliccp has only 3 scenarios while Cloud Theme has 355, it has two impacts: (1) scenario bias in Aliccp is smaller than in Cloud Theme. So $\alpha$, which represents the amount of scenario loss, will be smaller in Aliccp than in Cloud Theme. (2) There are more data in one scenario in Aliccp, so the interests are more concentrated in Aliccp. In this way, user interests have more influence on click behavior. As $c$ represents the counterfactual reference state of $M \rightarrow Y$ in Eq. (12), it is reasonable to be larger in Aliccp than in Cloud Theme.

## 5 RELATED WORKS

### 5.1 Single-scenario Recommendation

Most of the existing deep CTR models primarily concentrate on modeling a single scenario and follow the embedding and MLP paradigm. Wide&Deep [6] and DeepFM [9] combine the low-order (explicit interaction) and deep (implicit interaction) components to enhance performance. EDCN [4] further improves information sharing between different interaction networks in deep models through a parallel structure. DIEN [47] integrates the attention mechanism with GRU [7] to model the dynamic evolution of user interests over time. SIM [22] extracts user interests using two cascaded search units, enabling better modeling of lifelong behavior.

### 5.2 Multi-scenario Recommendation

As mentioned previously, the mainstream approach for addressing the multi-scenario problem is to create a unified framework that simultaneously models all scenarios. Consequently, our survey primarily focuses on works related to this paradigm. Specifically, Shared-bottom [3] constructs a shared bottom network to encode data from all scenarios, and different sub-networks to serve

different scenarios. MMoE [18] adopts a multi-gate mixture-of-experts technique to implicitly capture commonalities and distinctions among multiple scenarios. STAR [32] designs a star topology framework with a central network to capture overarching scenario commonalities and a set of scenario-specific networks to distinguish scenario-specific differences. The combination strategy of the element-wise product of layer weights serves as the information transfer mechanism from the overall scenarios to individual scenarios. PLE [34] divides experts in MMOE into scenario-shared experts and scenario-specific experts. AESM2 [49] adaptively selects suitable experts to obtain knowledge for the current scenario by calculating the distance between experts and scenarios. M2M [44] focuses on advertiser modeling in multiple scenarios and introduces a dynamic weights meta unit to model inter-scenario correlations. However, the aforementioned methods employ implicit approaches for scenario information transfer, making it challenging to explicitly represent the impacts of multiple scenarios. Furthermore, these models directly use click labels from other scenarios for training, ignoring the impact of scenario bias.

### 5.3 Causal Inference

Causal inference [21] is used in recommendation for debiasing [5], data missing, fairness, etc [16]. For example, IPS [30] adopted an inverse propensity weighting objective to learn unbiased matrix factorization models to address exposure bias. PD [46]introduced backdoor adjustment to remove the confounding popularity bias during model training and incorporated an inference strategy to leverage popularity bias. CR [39] and MACR [42] employ counterfactual inference to remove the effect of clickbait issues and popularity bias respectively. DCR-MOE [11] uses backdoor adjustment and designs an MOE structure network to address confounding features. CBDF [45] tackles the problem of data noise caused by delayed feedback with importance sampling to re-weight the original reward and obtain the modified reward in the counterfactual world. Though causal inference is widely employed to address various problems, scenario bias has not yet been addressed. We highlight the significance of this bias and propose the utilization of counterfactual inference as a means to mitigate its effects.

## 6 CONCLUSION

In this paper, we present M-scan, a model for multi-scenario recommendation systems. M-scan incorporates a Scenario-Aware Co-Attention mechanism to explicitly extract user interests from other scenarios that can match the current scenario at the feature level.. Additionally, M-scan includes a Scenario Bias Eliminator that employs causal counterfactual reasoning to mitigate biases introduced when we are using data from other scenarios to train models. M-scan demonstrates promising performance in offline experiments conducted on two public datasets, validating its effectiveness. The ablation experiments and hyperparameter analysis further confirm the utility of both modules. In future work, we plan to explore more advanced and comprehensive approaches to address scenario bias removal, continue designing more effective methods to leverage interests from other scenarios, delve into deeper causal reasoning techniques, and conduct online experiments. Finally, we may scale up multi-scenario models to large recommendation models.

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

# A EXPERIMENTAL SETTINGS

## A.1 Datasets

We conducted experiments using two publicly available multi-scenario datasets, which are listed in Table 3:

**Table 3: The data statistics.**

| Dataset | Users # | Items # | Scenarios # | Interactions # |
|---|---|---|---|---|
| Aliccp | 444,862 | 4,348,616 | 3 | 85,316,975 |
| Cloud Theme | 720,210 | 1,361,672 | 355 | 1,423,835 |

- **Aliccp [19]:** This dataset is provided by AliMama. It is collected from the recommendation system logs of the mobile Taobao application, including click data and associated conversion data. The dataset has 3 themes, which can be considered as a multi-scenario recommendation dataset.
- **Cloud Theme [8]:** This dataset contains user click logs from the cloud theme scenario in the Taobao app. It is used to optimize recommendations for users in multiple different scenarios. It consists of 355 scenarios and 1.4 million records.

**Train & test splitting**. We first filter out the users who own behaviors across multiple scenarios and then sort all the logged samples in chronological order. Finally, we split the most recent 40% samples as the test set while the other samples are put into the training set because we always use the old data to train and infer on the new data. It is a widely used splitting strategy in recommendation tasks. [23, 24]

## A.2 Baseline Model

The baseline models compared in the experiments are listed as follows:

- **Single**. The model is trained only with samples from the target scenario. Specifically, three-layer fully connected networks are applied for the experiments.
- **Mix**. We refer to the Mix as the model trained with a mixture of samples from all scenarios. The model structure is the same as the Single.
- **Finetune**. Finetune is a commonly-used and effective domain adaption (DA) training manner in industrial recommendation system [40, 41]. It first trains a unified model with the mixture of samples from all scenarios (namely the Mix), then adjusts the unified model with the data of the target scenario.
- **Shared bottom** [3]: Shared bottom is a widely used multi-scenario multi-task model that shares the parameters of the bottom network. Specifically, we use the embedding layer as the shared part and design 3 fully connected layers for each scenario on top of the shared part.
- **MMOE** [18]: MMOE is a multi-scenario multi-task model that is based on the shared bottom and uses multiple expert networks to learn knowledge for multiple scenarios and tasks, which are then integrated for prediction.
- **PLE** [34]: PLE is currently a state-of-the-art multi-scenario model. Compared to MMOE, it divides the experts into scenario-shared and scenario-specific ones and uses a progressive path mechanism to extract deep knowledge from experts.
- **AESM2** [49]: AESM2 is a state-of-the-art multi-scenario model that adaptively selects suitable experts to obtain knowledge for the current scenario by calculating the distance between experts and scenarios.
- **M2M** [44]: M2M is a state-of-the-art multi-scenario model that uses meta-learning techniques to design meta-attention and meta-network mechanisms for the multi-scenario framework, helping each scenario obtain its unique network.

