# OpenReview forum: "M-scan: A Multi-Scenario Causal-driven Adaptive Network for Recommendation"
_ACM.org/TheWebConf/2024/Conference — TheWebConf24_

### Official Review · Reviewer_WY5W · 2023-11-21

**Novelty:** 5
**Technical Quality:** 5

**Review:**

This work studies the multi-scenario recommendation task by proposing a causal-driven adaptive network. Compared with previous studies focusing on innovative model architectures, this work explores this task at the individual user level by leveraging the casual graph analysis. The overall solution sounds reasonable, and the results on the real-world datasets show improvements over other related methods.

Strong points
1. This work studies the multi-scenario recommendation task by proposing a causal-driven adaptive network to address the prediction bias introduced by data from other scenarios.
2. This proposal analyzes the impact of scenarios not only on user interests but also directly on click behavior using causal graphs.
3. The overall solution sounds reasonable, and the results on the real-world datasets show improvements over other related methods.

Weak points
1. The multi-scenario recommendation task is similar to the cross-domain recommendation (CR), but the authors do not give the differences between them and the related studies on CR tasks.
2. This work says that in the final inference, it is incorrect to consider the user’s interests across all scenarios, which is counterintuitive. More discussions and explanations on this are expected.
3. The work does not clarify its limitations, especially the scenarios in which the proposed method can be applied.

**Questions:**

Weak points 1-3.

**Reviewer Confidence:**

3: The reviewer is confident but not certain that the evaluation is correct

**Scope:**

4: The work is relevant to the Web and to the track, and is of broad interest to the community

---

### Official Review · Reviewer_jwnE · 2023-11-22

**Novelty:** 4
**Technical Quality:** 5

**Review:**

This paper aims to the recommendation system in multi scenarios. It uses a Scenario Bias Eliminator to reduce the influences of biases from other scenarios when considering the direct causality between the scenarios and the final click behaviors. The Scenario-Aware Co-Attention module is applied to explicitly extract the users' interests' from all scenarios. The experimental results proves the effectiveness of this model.

**Pros:**
1. The author notices both direct ($S \rightarrow Y$) and indirect ($S \rightarrow M \rightarrow Y$) influences of scenarios on the final user' behaviors and leverage them by eliminating the bias between scenarios.

**Cons:**
1. In model
   - The model seems to mismatch the manuscript. For example in Figure 3 the item embedding are fed into Scenario Encoder but it is not reflected in Section 3 (Equation 4). Does the $s_{b_k}$ means the calculation result of $S_u$ and $i$ ?
2. In experiment
   - To better prove the the first motivation (direct $S \rightarrow Y$) of M-scan, the experiments which remove the indirect causality (remove $S \rightarrow M$) should be conducted. The ablation w/ SACA, w/o SBE is close to this, but not able to remove the indirect causal influences.
3. In writing
   - L244: "u" should be in formula format.
   - Lack of punctuation after interline formula.
   - The last three paragraphs in Section 2.2 focus on the motivation of this paper, and should appear in Section 1 (Introduction) rather than Section 2 (Preliminaries).

**Questions:**

See above in review.

**Ethics Review Description:**

No  ethical issues.

**Reviewer Confidence:**

2: The reviewer is willing to defend the evaluation, but it is likely that the reviewer did not understand parts of the paper

**Scope:**

2: The connection to the Web is incidental, e.g., use of Web data or API

---

### Official Review · Reviewer_vouN · 2023-11-22

**Novelty:** 4
**Technical Quality:** 5

**Review:**

Summary
To address the bias in multi-scenario recommendations, this paper proposed the Multi-Scenario Causal-driven Adaptive Network, namely M-scan, from the perspective of causality. This method employs a Scenario Bias Eliminator module to debias information learned from other scenarios with counterfactual causality, and a Scenario-aware Co-attention mechanism to capture user interests that are consistent cross scenarios.

Strengths
S1: The proposed a novel and effective way to explicitly models the unique interest representations of multiple scenarios at the feature or data level.
S2: The authors conducted experiments on two public datasets and demonstrated the effectiveness of M-scan in offline experiments. The ablation experiments and hyperparameter analysis further confirmed the utility of both modules.

Weakness
W1: How the counterfactual reasoning is used in the Scenario Bias Eliminator module is not well demonstrated. For example, in section 3.3, the author states “counterfactual means considering S as s*, representing its removal from reality…”. In this sentence, what the counterfactual s* stands for and how we can get it is missing. Moreover, the idea of counterfactual reasoning is conducting minimal intervention on the treatment variable to observe a reverse outcome. The “minimal intervention” and “reverse outcome” are not reflected in the proposed method.

W2: The overall writing is not clear enough. For example, at the end of the Introduction section, the author argues that the M-scan is the first to analyze the impact of scenarios not only on user interests but also directly on click behavior using causal graphs. I am not sure if the author targets the task of cross-domain recommendation or multi-behavior recommendation. However, there is already a load of causal-based models proposed in both of the tasks to mitigate different biases including but not limited to popularity bias or to capture the consistent interest of users for better generalization.

W3: From Figure 2 to Figure 4, the Scenario Identifier (denoted by S) is decomposed to S and S*. However, when doing this decomposition, the factual distribution of Y will be influenced. This contradicts the idea of counterfactual reasoning. The proposed debiasing module is not convincible, especially with the availability of traditional deconfounding methods.

W4: There are multiple notation errors in the formulas. For example, in Eq. (12), what is *?

**Questions:**

Q1: Why the proposed M-scan is superior to traditional deconfounding methods such as inverse propensity scoring and backdoor adjustment?

**Reviewer Confidence:**

4: The reviewer is certain that the evaluation is correct and very familiar with the relevant literature

**Scope:**

4: The work is relevant to the Web and to the track, and is of broad interest to the community

---

### Official Review · Reviewer_nTRt · 2023-11-25

**Novelty:** 5
**Technical Quality:** 5

**Review:**

In summary, the paper presents a technically sound and innovative approach to tackling the inherent challenges in multi-scenario recommendation systems.

**Quality and clarity**

The paper is well-structured, presenting the problem, methodology, and solution in a coherent manner. The paper dives deep into the technical aspects, particularly the causal-driven analysis and the introduction of the M-scan model. The use of causal graphs for understanding the multi-scenario recommendation system is well explained.



**Originality**

The paper introduces a new approach with its Scenario-Aware Co-Attention and Scenario Bias Eliminator modules, which appear to be novel contributions to the field of recommendation systems.
The authors highlight their work as the first paper that analyzes the impact of scenarios not only on user interests but also directly on click behavior using causal graphs.



**Pros**

1. The model addresses key challenges in multi-scenario recommendation systems, such as data sparsity in some scenarios and bias due to direct incorporation of data from various scenarios.
2. The use of causal counterfactual inference to mitigate biases is a significant strength, offering a potentially more robust model.



**Cons**

1. **Lack of Detailed Experimental Settings**: The paper does not provide a comprehensive description of the experimental settings, such as the basic learning rate and number of epochs. This omission makes it difficult to fully evaluate the experimental methodology and replicate the study.
2. **Limited Dataset Analysis**: My main concern is: the main results, as shown in Table 1, are derived from only two datasets. This limited dataset analysis restricts the comprehensiveness of the results. It would be beneficial for the authors to include additional datasets and metrics to validate the robustness and generalizability of their model across different scenarios and data environments.

3. **Section on Related Works Could Be Enhanced**: Section 5.3, which discusses related works, may be better titled as 'Causal Inference for Recommender Systems' to more accurately reflect the content and focus of the section.
Some previous works have leveraged causality to analyze the biases in RS.
To strengthen the paper, the authors might consider summarizing this research line, claiming your distinct role (like consider a different bias) and incorporating the following works: Recommendations as treatments: Debiasing learning and evaluation; Causal embeddings for recommendation; Causal inference for recommender systems; Contrastive Counterfactual Learning for Causality-aware Interpretable Recommender Systems.

I will also check the comments from other reviewers and would like to increase the score if the authors solve the above issues.

**Questions:**

See cons.

**Reviewer Confidence:**

3: The reviewer is confident but not certain that the evaluation is correct

**Scope:**

3: The work is somewhat relevant to the Web and to the track, and is of narrow interest to a sub-community

---

### Decision · Program_Chairs · 2024-01-22

**Decision:**

Accept

**Comment:**

This paper introduces the Multi-Scenario Causal-driven Adaptive Network with a focus on causality. Reviewers acknowledge the novelty of the work and the sufficiency of its experiments. To enhance the paper, it is recommended that the authors include the information from their rebuttal to further strengthen their research.